# Long-Term Impacts of COVID-19 Pneumonia on Quality of Life: A Single Institutional Pilot Study

**DOI:** 10.3390/healthcare11131963

**Published:** 2023-07-07

**Authors:** Athavudh Deesomchok, Chalerm Liwsrisakun, Warawut Chaiwong, Chaicharn Pothirat, Pilaiporn Duangjit, Chaiwat Bumroongkit, Theerakorn Theerakittikul, Atikun Limsukon, Pattraporn Tajarernmuang, Konlawij Trongtrakul, Nutchanok Niyatiwatchanchai

**Affiliations:** Division of Pulmonary, Critical Care and Allergy, Department of Internal Medicine, Faculty of Medicine, Chiang Mai University, Chiang Mai 50200, Thailand

**Keywords:** quality of life, COVID-19 pneumonia, long-term follow-up, anxiety, depression

## Abstract

Many studies have demonstrated poor quality of life (QoL) at 3, 6, 12, and 24 months after coronavirus disease 2019 (COVID-19). However, these studies were limited due to cross-sectional design, a longer gap between visits, and lack of controls for comparison. Therefore, the aim of our prospective study was to assess the impact of COVID-19 pneumonia on QoL in both physical and mental health. A prospective study was conducted on adult patients with COVID-19 pneumonia. We used the 36-Item Short Form Health Survey (SF-36) and Euro Quality of Life-5 Dimensions-5 Levels (EQ-5D-5L), EQ visual analogue scale (EQ-VAS), and Hospital Anxiety and Depression Scale to collect data at months, 1, 3, 6, 9, and 12. Thirty-eight patients with COVID-19 pneumonia and twenty-five healthy subjects were completely followed up on all visits. All domains of SF-36, except bodily pain and EQ-5D-5L of the patients, were lower than controls. There was an improvement of EQ-VAS and SF-36 including physical functioning, social functioning, and role limitation (physical problems) domains throughout study period in the COVID-19 pneumonia group. Adult patients who recovered from COVID-19 pneumonia had lower QoL which improved over the one-year follow-up period.

## 1. Introduction

The coronavirus disease 2019 (COVID-19) infectious respiratory illness caused by SARS-CoV-2 has been reported since late 2019, led to a pandemic, and has severely impacted human health worldwide [1]. COVID-19 infected more than 700 million people and led to over 6-million deaths worldwide until February 2023 [1]. This infection involves mainly the respiratory system [2,3]. In unvaccinated athletes, infection with SARS-CoV-2 resulted in decline of respiratory muscle strength, both maximal inspiratory pressure (MIP) and maximal expiratory pressure (MEP) [4]. They also found that COVID-19 affected pulmonary function, including forced expiratory volume in the first second (FEV_1_), forced vital capacity (FVC), and peak expiratory flow (PEF) [4].

Many patients experienced long-term physical and mental health problems called “post COVID-19 condition” [5] or “Long COVID” [6,7,8]. Our previous study reported that nearly one half of patients with post-COVID-19 pneumonia still had symptoms, including poor quality of life (QoL) and limited exercise capacity compared to the healthy subjects, one month after hospital discharge [9]. Health-related QoL measured by Euro Quality of Life-5 Dimensions-5 Levels (EQ-5D-5L) was significantly lower in the post severe COVID-19 pneumonia group compared to the post non-severe COVID-19 pneumonia and healthy control groups [9]. When assessing QoL measured by the 36-Item Short form Health Survey (SF-36), we discovered that physical, emotional, social, and mental health symptoms were significantly lower in the post-COVID-19 pneumonia group compared to healthy controls [9]. Exercise capacity measured by six-minute walk distance (6-MWD) was also significantly lower in the post severe COVID-19 pneumonia group compared to the post non-severe COVID-19 pneumonia and healthy control groups [9]. Additionally, previous studies demonstrated poorer health-related QoL at 3, 6, 12, and 24 months after COVID-19 infection [10,11,12,13,14]. For example, Todt et al. found that the Euro Quality of Life-5 Dimensions-3 Levels (EQ-5D-3L) at 3 months was significantly lower compared to before the onset of COVID-19 infection [10]. Huang et al. also found that health-related QoL measured using EQ-5D-5L continued to improve in almost all domains throughout study periods (6 months, 12 months, and 24 months) [13]. However, the results from these studies may be limited by the cross-section design, longitudinal and long interval between the visits, and lack of control group [10,11,12,13,14]. Therefore, this study aimed to explore the long-term impacts on health-related QoL in post-COVID-19 pneumonia patients compared with healthy control subjects with a short interval between visits (months 1, 3, 6, 9, and 12 after hospital discharge).

## 2. Materials and Methods

### 2.1. Study Design

This prospective observational study was conducted in an outpatient clinic at the Lung Health Center, Division of Pulmonary, Critical Care, and Allergy, Department of Internal Medicine, Faculty of Medicine, Chiang Mai University, Chiang Mai, Thailand, between May 2021 and April 2022. This study was approved by the Research Ethics Committee, Faculty of Medicine, Chiang Mai University (Study code: MED-2564-08109, date of approval: 3 May 2021) and by the Clinical Trials Registry (Study ID: TCTR20210827005, date of approval: 27 August 2021) in compliance with the Declaration of Helsinki. All subjects provided written informed consent prior to enrollment. We included subjects ≥ 18 years of age with the diagnosis of COVID-19 pneumonia with clinical symptoms and evidence of pulmonary infiltration on chest X-ray (CXR) and a positive reverse transcription-polymerase chain reaction (RT-PCR) between April and May 2021. Only subjects with first-time infection with Alpha variant of SARS-CoV-2 were included in the study. We excluded subjects with language barrier (understanding Thai), diagnosis of psychiatric disorders (depression and/or anxiety), chronic obstructive pulmonary disease (COPD), and asthma. Healthy control subjects with age- and sex-matched were recruited for comparison.

### 2.2. Data Collection

Baseline characteristics such as age, sex, body mass index (BMI), underlying diseases, smoking status, complete blood count (CBC), vital signs, and severity of COVID-19 pneumonia during admission were collected at the first visit (one month after hospital discharge). History of COVID-19 vaccination and COVID-19 infection during study period were also collected. At each visit, clinical symptoms and signs including fatigue, dyspnea, chest pain, cough, headache, bodily pain, anosmia, cognitive dysfunction, insomnia, myalgia, and diarrhea; QoL questionnaires including the SF-36 and the EQ-5D-5L; and the Hospital Anxiety and Depression Scale (HADS) were collected as described in our previous study [9].

The SF-36 questionnaire developed by the research and development (RAND) Corporation, measures general health status in eight domains with a total of 36 questions including physical, social mental, emotional, and just general health perception. The summation of scores from all questions within each domain is calculated on a scale of 0 “worst health” to 100 “best health” [15]. At the last visit, the physical component summary scores (PCS) and mental component summary scores (MCS) from SF-36 questionnaire of the post-COVID-19 pneumonia group were calculated by using the method developed by Ware et al. [16]. The z-score standardizations for each component of SF-36 were computed by using mean and standard deviation (SD) of the healthy control group.

The EQ-5D-5L questionnaire developed by the EuroQol Research Foundation measures health status. It comprises the EQ-5D questionnaire and the EQ visual analogue scale (EQ-VAS). The EQ-5D questionnaire provides a descriptive health profile from no problem to slight, moderate, severe, and unable to/extreme problems (5 levels) of five dimensions: mobility, self-care, usual activities, pain/discomfort, and anxiety/depression. The summary index value is calculated from each level of these dimensions by using a standard EQ-5D-5L value set for Thailand. An index value close to 1.000 defines better QoL. The EQ-VAS is overall self-rated current health status on a vertical scale from 0 “The worst health you can imagine” to 100 “The best health you can imagine” [17].

The HADS questionnaire, developed by Zigmond and Snaith, assesses anxiety and depression in people one week prior to questionnaire response. It comprises seven questions for anxiety and seven questions for depression, each question is scored from 0–3 giving a total score ranging from 0–21 for anxiety (HADS-A) or depression (HADS-D). A HADS-A or HADS-D score ≥ 11 indicates the probable presence of anxiety or depression [18].

The patients were followed up at months 1, 3, 6, 9, and 12 after discharge from Maharaj Nakorn Chiang Mai Hospital, Chiang Mai, Thailand.

### 2.3. Study Size Estimation

Sample size calculation was based on the mean and SD of EQ-5D-5L one month after hospital discharge between post severe COVID-19 pneumonia group and control group in the previous study published by Niyatiwatchanchai et al. [9]. The means and SD of EQ-5D-5L in the post severe COVID-19 pneumonia group and the group with healthy control subjects were 0.77 ± 0.17 and 0.89 ± 0.12, respectively. The calculated effect size was 0.82. We calculated the number of participants to be 50 (25 in each group) to see if hypothesized effects that the population means between groups were equal with a probability (power) of 0.8. The type I error probability associated with this test of the null hypothesis was 0.05.

### 2.4. Statistical Analysis

Continuous data were expressed as mean and SD or standard error of means (SEM), whereas categorical data were expressed as numbers and percentages. The independent sample *t*-test or Mann–Whitney U test was used to compare differences between the two groups for continuous data and non-parametric data, respectively, whereas Fisher’s exact test was used to compare the categorical data between groups. Mixed repeated-measures analysis of variance (ANOVA) with Bonferroni correction was used for comparison numerical data within groups (5 time-points) and between groups throughout the study period. A *p* Value less than 0.05 was considered statistically significant. All statistical analyses were performed using STATA version 16 (StataCorp, College Station, TX, USA).

## 3. Results

Fifty-six patients from the post-COVID-19 pneumonia group and 25 subjects from the control group participated in the one-month post-hospital discharge follow up. Eighteen post-COVID-19 pneumonia patients (32.1%) withdrew or dropped out for various reasons (i.e., immigration, loss of contact); therefore, only 38 patients completed the follow-up visit. The 38 remaining patients with 20 male (52.6%) and mean age of 41.1 ± 14.8 years were included for analysis. Twenty-two (57.9%) of them had severe pneumonia, defined by COVID-19 pneumonia that required treatment with high flow nasal cannula (HFNC) or mechanical ventilator (MV) during admission. The mean age, proportion of male sex, underlying diseases, hematological data, and vital signs were comparable between the post-COVID-19 pneumonia and the healthy control groups, except for platelet count and pulse rate which were significantly higher in post-COVID-19 pneumonia group (337.9 ± 115.4 × 10^3^/mm^3^ vs. 273.0 ± 74.5 × 10^3^/mm^3^, *p* Value = 0.016 and 94.9 ± 12.1 beats/min vs. 83.8 ± 13.7 beats/min, *p* Value = 0.001, respectively). However, the post-COVID-19 pneumonia group had significantly lower oxygen saturation via pulse oximeter (SpO_2_) (97.2 ± 1.6% vs. 98.2 ± 0.9%, *p* Value = 0.006). Moreover, the post-COVID-19 pneumonia group had significantly higher BMI and more history of smoking than the healthy control subjects. More data on the details of demographic data are presented in Table 1.

During the study period, 36 patients in the post-COVID-19 pneumonia arm and everyone in the control arm received at least one dose of COVID-19 vaccination. Eight patients in the study arm and five in the control arm had re-infection with SARS-CoV-2. All reinfected patients had mild symptoms. More data are shown in Table 1.

During a follow-up visit, the post-COVID-19 pneumonia patients had at least one symptom in 57.9%, 34.2%, 28.9%, 34.2%, and 44.7% at 1-month, 3-month, 6-month, 9-month, and 12-month visits, respectively. The number of patients with at least one symptom was significantly higher than healthy control subjects at 1-month and 3-month visit (*p* Value < 0.001 and *p* Value = 0.018, respectively). From month 6 to month 12, the number of patients with at least one symptom was also higher than healthy control subjects but not significant. Cough and fatigue were the most common symptoms (Figure 1 and Figure 2). The overall symptoms resolved over time but rebounded at months 9 and 12 (Figure 1). A similar trend was observed in patients who only had cough symptoms (Figure 2).

Health-related QoL, measured by EQ-5D-5L index value, in post-COVID-19 pneumonia patients, was significantly lower than healthy control subjects throughout the study period (*p* Value = 0.007) and in every follow-up visit. In the post-COVID-19 pneumonia group, the improvement of EQ-5D-5L was not significantly improved throughout the study period (*p* Value = 0.156). However, the EQ-5D-5L improved from month 1 to month 9 and then reached a plateau from month 9 to month 12. The EQ-VAS in the post-COVID-19 pneumonia group tended to be lower than healthy controls throughout the study period (*p* Value = 0.058). However, the EQ-VAS was significantly lower in post-COVID-19 pneumonia patients at month 1 and month 3. The scale increased throughout the study period in the post-COVID-19 pneumonia group (*p* Value = 0.042). It was significantly improved after the 6-month visit when compared with the 1-month visit. The details of the health-related QoL of both groups are presented in Table 2.

Although the SF-36-derived QoL progressively improved in post-COVID-19 pneumonia group, it was significantly lower than healthy control group throughout the study period in all domains except for bodily pain. Role limitations due to physical and social functioning domains in post-COVID-19 pneumonia patients were significantly improved throughout the study period. Three domains of SF-36, including physical functioning, social functioning, and role limitations physical problems were significantly improved at month 9 and month 12 visits when compared to month 1 visit. More data are shown in Figure 3. At 12 months, the PCS and MCS scores of the study group were 40.8 and 45.0, respectively.

There was no significant difference between anxiety and depression between post-COVID-19 pneumonia patients and healthy control subjects. A few post-COVID-19 pneumonia patients had an anxiety, defined by HADS-A score ≥ 11, [1 (2.6%), 1 (2.6%), 2 (5.3%), 3 (7.9%) and 1 (2.6%) patients] or depression, defined by HADS-D score ≥ 11, [4 (7.9%), 1 (2.6%), 2 (5.3%), 1 (2.6%) and 1 (2.6%) patients] at months 1, 3, 6, 9, and 12, respectively (Figure 4). No healthy control subjects had anxiety or depression.

## 4. Discussion

The long-term impact of COVID-19 pneumonia on health-related QoL and psychological problems were consecutively measured from month 1, 3, and every 3 months till 12 months after hospitalization for COVID-19 pneumonia. Our study demonstrated that QoL was lower in patients with post-COVID-19 pneumonia compared to the control group. In addition, despite gradual improvement during the entire follow-up period, it remained lower than the control group.

A proportion of our patients still had at least one symptom during follow-up visits. Similar to the study by Huang et al. which demonstrated that 68% and 49% of post-COVID-19 patients had at least one symptom at 6 and 12 months, respectively [12]. We found that the number of patients with at least one symptom was initially higher than healthy control subjects. However, it became insignificant after month 3. At 6 and 12 months, only 34.2% and 44.7% of our patients still had at least one symptom. The reason for the lower rate of symptoms than the study by Huang et al. might be from younger age group in our cohort. We also demonstrated that cough and fatigue were the most common symptoms in the study group and presented at a significantly higher rate than in the control group. The most persistent symptom during later visits was cough. This finding was different from study by Huang et al. who showed that fatigue was the most common symptom at 6 and 12 months [12]. The cause was Huang et al. did not report the cough symptoms of their subjects during first year after recovery from COVID-19 in their study. Furthermore, our subjects had more severe COVID-19 because the number of patients who required HFNC or MV in our study and in their study were 57.9% vs. 7.6%, respectively. The importance of cough symptom was confirmed by our previous study which showed that cough was the most common symptom at one month post-COVID-19 pneumonia, especially in severe disease, which might be caused by the residual lung lesion after pneumonia [9]. Fumagalli et al. performed a serial telephone follow-up study in previously hospitalized patients with COVID pneumonia for a 12-month duration like our study. They reported that cough was a common long COVID symptom and their prevalence of cough at months 3 and 6 were equivalent to our study [19]. In contrast, their prevalence of cough became progressively lower as time went by [19]. Therefore, the pattern of cough prevalence in our study was unusual. Our study recruited patients during April and May 2021. At 9 months and 12 months after enrollment, it would be between December 2021 and April 2022 when the air pollution, particulate matter with a diameter of smaller than 2.5 microns (PM_2.5_), in Chiang Mai was higher than the standard World Health Organization (WHO) level annually [20,21,22,23]. The rebounding cough in the present study was the most likely effect of PM_2.5_-induced airway inflammation. In addition to chronic cough, fatigue was another common long COVID-19 symptom. As described earlier that COVID-19 is a systemic disease, chronic inflammation of the brain and neuromuscular junctions, sarcolemma damage and fiber atrophy and damage, and psychosocial factors might contribute to post-COVID-19 fatigue [7]. Both residual symptoms could affect their health-related QoL.

SF-36 and EQ-5D-5L are tools for assessing general health-related QoL in various domains including physical function and mood disorders [15,17]. Our study revealed that the SF-36 and the EQ-5D-5L in post-COVID-19 pneumonia patients were worse than healthy subjects during follow-up visits which were supported by the previous studies [10,11,12]. Our study also showed that EQ-5D-5L index value in post-COVID-19 pneumonia patients was significantly lower than healthy control subjects throughout the study period and in every follow-up visit. Moreover, we found that the EQ-5D descriptive system and EQ-VAS of these patients improved, with variable significance, from month 1 to month 6 and then stable in later visits. For the SF-36 QoL questionnaire, role limitations due to physical and social functioning domains in post-COVID-19 pneumonia patients were significantly improved throughout the study period. Most of the domain of social functioning and role limitations in physical problems at the 9-month and 12-month visits were significantly better than at 1-month follow-up, whereas the bodily pain was the only domain that was not different from healthy control subjects and did not improve over time. Like the study by Deana et al. [24], our study showed that post-COVID-19 pneumonia affected global physical wellness more than mental wellness as shown by lower PCS score than MCS score at 12 months. These findings might be explained by the disease itself (post-viral syndrome or viral-associated organ damage) and/or non-COVID-19 psychosocial factors such as poor sleep quality, reduced physical activities, fear, anxiety, depression, and economic problems [25]. Therefore, the COVID-19-specific questionnaire is needed to explore the mechanism of post-COVID-19 physical involvement in the future studies.

Only a few of our post-COVID-19 pneumonia patients had anxiety or depression assessed by HADS questionnaire, which was less prevalent than in other studies (2.6% vs. 21.0% and 2.6% vs. 19.0% at months 3 and 12 for anxiety, 2.6% vs. 12.0% and 2.6% vs. 15.0% at month 3 and 12 for depression [14]; 10.6% vs. 23.0% and 5.3% vs. 26.0% at month 6 and 12 for anxiety or depression [12]. The discrepancy related to the prevalence of depression and anxiety between our study and previous studies may be due to the age of the study population. The median age of participants from the studies of Lorent et al. and Huang et al. was 59 years old which is higher than the mean age of our patients (41 years old) [12,14]. A previous study found that increasing age was associated with a higher risk of anxiety or depression [26]. Not only aging, the direct effects of COVID-19, underlying medical illnesses, sociodemographic factors (e.g., female, living alone), psychosocial factors (e.g., poor self-rated health, poor sleep quality, less support from family and society, high social media exposure), and job-related factors also contributed to psychiatric symptoms in COVID-19 patients and general population during COVID-19 pandemic [27].

The strengths of our study were the prospective design, it being 12-month longitudinal study, and the inclusion of control group. This study had several limitations. Firstly, this was a single tertiary center trial. Thus, the results may not be generalized for the other clinical setting. Secondly, nearly one-third of post-COVID-19 pneumonia patients dropped out during follow-up visits. Therefore, there was a selection bias in our study. Thirdly, additional confounding factors such as air pollution [20,21] and the new variant [28] during the 9th month (January–February) and 12th month visit (April–May 2022) might have influence on our results. Fourthly, due to a small sample size of severe (N = 22) and non-severe (N = 16) COVID-19 pneumonia, the differences in health related QoL between severe and non-severe COVID-19 pneumonia groups were not compared in our study. Thus, it should be mentioned in future studies. Fifthly, the results from SF-36 were not correlated with the clinical relevance. Thus, the association between SF-36 and clinical correlation should be accounted for in future investigation.

## 5. Conclusions

The sequelae of post-COVID-19 pneumonia on health-related QoL were observed in our study. Among patients with post-COVID-19 pneumonia, one-third of them still had symptoms, mainly cough and fatigue, after 6 months of recovery from the disease. They also had lower health-related QoL which expected to improve over time during 1 year of follow-up. Therefore, a multidisciplinary program including physical rehabilitation and psychosocial support is required for managing patients with ongoing complications post-COVID-19 infection.

## Figures and Tables

**Figure 1 healthcare-11-01963-f001:**
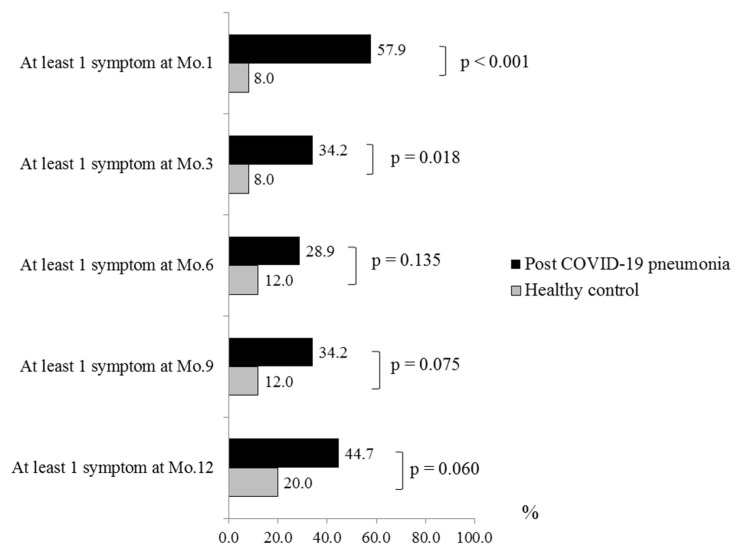
The long-term impact of post-COVID-19 pneumonia on symptoms. Abbreviation: COVID-19, coronavirus disease 2019.

**Figure 2 healthcare-11-01963-f002:**
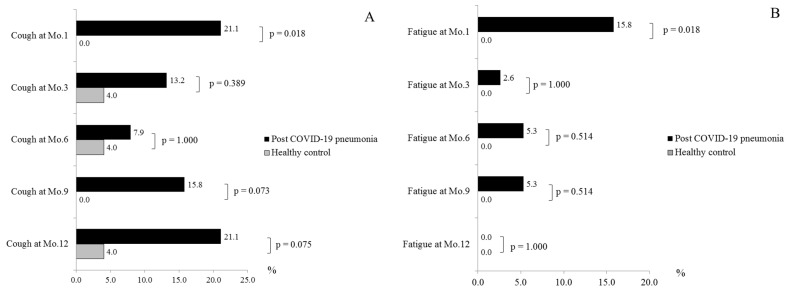
The Long-Term Impact of Post-COVID-19 Pneumonia on Symptoms of Cough (**A**) and Fatigue (**B**). Abbreviation: COVID-19, coronavirus disease 2019.

**Figure 3 healthcare-11-01963-f003:**
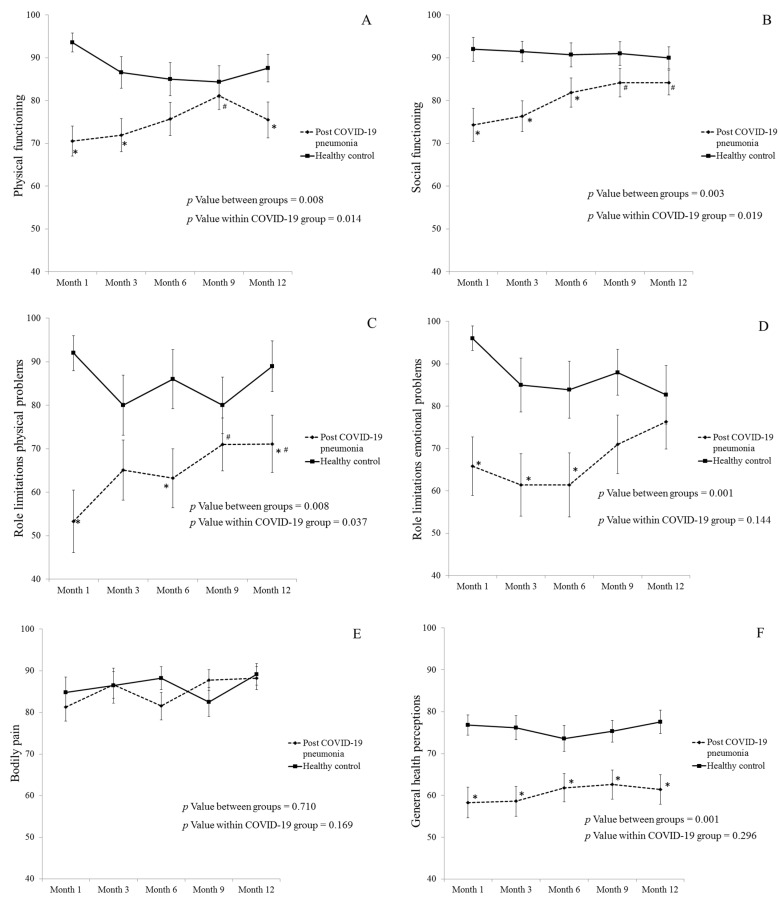
Health-Related Quality of Life Measured by SF-36 during follow-up period in post-COVID-19 pneumonia patients compared to healthy controls. Physical functioning (**A**), social functioning (**B**), role limitations physical problems (**C**), role limitations emotional problems (**D**), bodily pain (**E**), general health perceptions (**F**), mental health (**G**), and vitality (**H**). Note: Data are presented as mean ± SEM; * *p* Value < 0.05 compared to healthy controls; ^#^
*p* Value < 0.05 compared to month 1 visit. Abbreviations: COVID-19, coronavirus disease 2019; SF-36, 36-item Short-Form Health Survey.

**Figure 4 healthcare-11-01963-f004:**
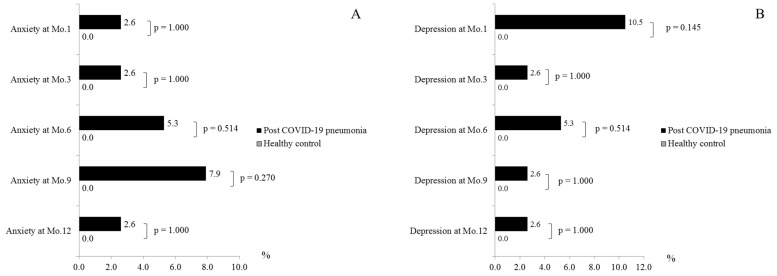
Hospital Anxiety and Depression Scale Measured by HADS Questionnaire in post-COVID-19 pneumonia patients compared to healthy controls. Anxiety (**A**) and Depression (**B**). Abbreviation: HADS, the Hospital and Anxiety and Depression Scale questionnaire; COVID-19, coronavirus disease 2019.

**Table 1 healthcare-11-01963-t001:** Baseline Characteristics of Study Population (n = 63).

Variables	Post-COVID-19 Pneumonia (n = 38)	Healthy Control Subjects (n = 25)	*p* Value
Age (years)	41.1 ± 14.8	43.0 ± 9.6	0.555
Male sex, n (%)	20 (52.6)	12 (48.0)	0.961
Body mass index (kg/m^2^)	29.0 ± 5.1	26.1 ± 5.6	0.036
Underlying diseases			0.588
Cardiovascular	9 (23.7)	5 (20.0)	
Metabolic	1 (2.6)	0 (0.0)	
Cardiovascular + metabolic	4 (10.5)	1 (4.0)	
None	24 (63.2)	19 (76.0)	
Smoking status			0.039
Current	1 (2.6)	4 (16.0)	
Ex-smoker	8 (21.1)	1 (4.0)	
Non-smoker	29 (76.3)	20 (80.0)	
Complete blood count			
Hemoglobin (g/dL)	13.2 ± 1.5	13.2 ± 1.7	0.911
Hematocrit (%)	39.7 ± 3.9	39.8 ± 3.9	0.882
White blood count (×10^3^ cells/mm^3^)	8.7 ± 5.9	6.9 ± 1.9	0.148
Neutrophil count (×10^3^ cells/mm^3^)	5.0 ± 3.6	4.0 ± 1.4	0.192
Lymphocyte count (×10^3^ cells/mm^3^)	2.9 ± 1.9	2.3 ± 0.5	0.142
Eosinophil count (cells/mm^3^)	166.7 ± 140.4	227.6 ± 252.5	0.227
Platelet count (×10^3^/mm^3^)	337.9 ± 115.4	273.0 ± 74.5	0.016
Vital signs			
Temperature (°C)	36.6 ± 0.3	36.5 ± 0.3	0.155
Pulse rate (beats/min)	94.9 ± 12.1	83.8 ± 13.7	0.001
Respiratory rate (breaths/min)	18.9 ± 1.1	18.3 ± 2.2	0.204
Mean arterial pressure (mmHg)	102.0 ± 9.1	96.6 ± 14.2	0.101
SpO_2_ (%)	97.2 ± 1.6	98.2 ± 0.9	0.006
Severity of COVID-pneumonia during admission			
Non-severe (no O_2_ therapy or required low-flow O_2_ cannula)	16 (42.1)	N.A.	
Severe (required HFNC or MV)	22 (57.9)	N.A.	
Received at least one dose of COVID-19 vaccine	36 (94.7)	25 (100.0)	1.000
Re-infection with SARS-CoV-2 during study period	8 (21.1)	5 (20.0)	0.514

Note: Data are mean ± SD or n (%). Abbreviation: SpO_2_, oxygen saturation via pulse oximeter; HFNC, high flow nasal cannula oxygen; MV, mechanical ventilation; COVID-19, coronavirus disease 2019; O_2_, oxygen saturation.

**Table 2 healthcare-11-01963-t002:** Health-Related Quality of Life Measured by EQ-5D-5L during Follow-up Period.

Follow Up Period	EQ-5D-5L Index Value (0–1)		EQ-VAS (0–100)	
	Post-COVID-19 Pneumonia (n = 38)	Healthy Control Subjects (n = 25)	Post-COVID-19 Pneumonia (n = 38)	Healthy Control Subjects (n = 25)
Month 1	0.78 ± 0.18 *	0.89 ± 0.12	81.8 ± 11.6 *	87.4 ± 9.5
Month 3	0.79 ± 0.17 *	0.88 ± 0.13	83.6 ± 10.0 *	88.6 ± 8.8
Month 6	0.79 ± 0.16 *	0.89 ± 0.12	85.9 ± 9.7 ^#^	88.4 ± 8.6
Month 9	0.82 ± 0.17 *	0.89 ± 0.14	86.2 ± 10.5 ^#^	89.6 ± 7.8
Month 12	0.82 ± 0.17 *	0.92 ± 0.12	85.9 ± 10.7 ^#^	89.9 ± 7.6
*p* Value within group **	0.156	0.073	0.042	0.291
*p* Value betwwen groups **	0.007		0.058	

Note: Data are presented as mean ± SD; * *p* Value < 0.05 compared to healthy controls; ^#^
*p* Value < 0.05 compared to visit month 1; **, *p* Value from repeated-measures analysis of variance (ANOVA). Abbreviations: EQ-5D-5L, Euro Quality of Life-5 Dimentions-5 Levels; EQ VAS, Euro Quality of Life visual analogue scale; COVID-19, coronavirus disease 2019.

## Data Availability

The datasets used and/or analyzed during the current study are available from the corresponding author upon reasonable request.

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
