# Peer review of "Long-Term Impacts of COVID-19 Pneumonia on Quality of Life: A Single Institutional Pilot Study"

_healthcare, 2023, doi:10.3390/healthcare11131963_

Round 1

Reviewer 1 Report

Comments to the Author

The authors of this article did an admirable job on an important topic, aimed to to explore the long-term impacts on health-related quality of life in post COVID-19 pneumonia patients (months 1, 3, 6, 9, and 12 after hospital discharge). This paper is well-organized, pertinent, and may add to the literature base of an important. However, there are several points that require further clarity;

1- Page 1, Line 31: It is worth mentioning that studies have demonstrated reduced pulmonary function as a result of COVID-19, as the virus primarily targets the respiratory system and diminishes lung capacities. Consequently, these adverse physiological effects may lead to a decline in the quality of life and exercise capacity of individuals. Talking of exercise, you can add a few studies on athlete populations. Here are a few sample studies for your reference.

https://doi.org/10.1016/j.resp.2022.103983

https://doi.org/10.3390/ijerph18084065

2- Page 1, Lines 37-61: Please remove numerical and statistical data from this section.

3- Page 2, Line 63: It would also be good to provide information on the number of times the subjects had COVID-19 and the variant available in the country at that time. It would also be useful to indicate whether they were reinfected at the time of follow-up.

4- Page 3, Line 125: It would have been better if mixed repeated-measures analysis of variance (RM ANOVA) (2x5) was used as a statistical method so that we could see the interaction effects (groupxtime).

GENERAL COMMENTS:

1. The topic is important but especially the introduction and discussion sections should be improved significantly. Literature review is nonadequacy.

2. More professional visualization applications should be used for graphics.

The language is good but needs a little editing.

Author Response

Date 20th June 2023

Editor-in-Chief

Healthcare Journal

Dear Reviewer 1

Subject:  Response to reviewer 1

            On behalf of my co-authors, I am pleased to resubmit this revised Manuscript ID: healthcare-2430660 entitled: “Long-Term Impacts of COVID-19 Pneumonia on Quality of Life” as attached to the Healthcare Journal.

Thank you for your valuable comments.

Comments and Suggestions for Authors

Comments to the Author

            The authors of this article did an admirable job on an important topic, aimed to to explore the long-term impacts on health-related quality of life in post COVID-19 pneumonia patients (months 1, 3, 6, 9, and 12 after hospital discharge). This paper is well-organized, pertinent, and may add to the literature base of an important. However, there are several points that require further clarity;

1- Page 1, Line 31: It is worth mentioning that studies have demonstrated reduced pulmonary function as a result of COVID-19, as the virus primarily targets the respiratory system and diminishes lung capacities. Consequently, these adverse physiological effects may lead to a decline in the quality of life and exercise capacity of individuals. Talking of exercise, you can add a few studies on athlete populations. Here are a few sample studies for your reference.

https://doi.org/10.1016/j.resp.2022.103983

https://doi.org/10.3390/ijerph18084065

Response: We added reference per your suggestion. Revised manuscript in the introduction section page no.1 line 30-34, highlighted and in reference section page no. 11, line 344-346, highlighted.

2- Page 1, Lines 37-61: Please remove numerical and statistical data from this section.

Response: We deleted numerical and statistical data from introduction section per your suggestion. Revised manuscript in the introduction section page no.1-2 line 35-47, highlighted.

3- Page 2, Line 63: It would also be good to provide information on the number of times the subjects had COVID-19 and the variant available in the country at that time. It would also be useful to indicate whether they were reinfected at the time of follow-up.

Response: We added this information in the study design sub section per your suggestion. Revised manuscript in the materials and method section page no.2 line 71-72, highlighted. Subjects with reinfected at the time of follow-up were mentioned in the table, page no.4, highlighted.

4- Page 3, Line 125: It would have been better if mixed repeated-measures analysis of variance (RM ANOVA) (2x5) was used as a statistical method so that we could see the interaction effects (groupxtime).

Response: We re-analyze data according to your suggestion and we added this test used in the statistical analysis part. Revised manuscript in the materials and method section page no.3 line 125-127, highlighted. The p-value from repeated measured ANONA was added in updated table 2 and figure 3. Revised manuscript was highlighted in the results section page no.6-8.

GENERAL COMMENTS:

  1. The topic is important but especially the introduction and discussion sections should be improved significantly. Literature review is nonadequacy.

Response: We re-write the introduction and discussion part per your suggestion. Revised manuscript in the introduction part page no.1, line 26-58, highlighted and discussion part page no. 8-10, line 225-243 and 260-267, highlighted.

  1. More professional visualization applications should be used for graphics.

Response: All figures were modified per your suggestion. Revised manuscript in the Figure 1-4, page no. 5, 7, and 8. 

Comments on the Quality of English Language

The language is good but needs a little editing.

Response: Thank you. This manuscript was sent to native English speaker for proving.           

With regards,

Assoc.Prof. Chalerm Liwsrisakun, MD.

Reviewer 2 Report

The authors have attempted to address an important issue related to post covid clinical and mental health consequences (QOL) in their Thai cohort. This appear to be a "follow on" reporting from their original study evaluating very early outcomes. They recruited a very small sample size based on their effects size -power analysis on one month post infection using their pre-existing outcomes based on the EQ -5D-5L survey . They report that 32% of the COVID group dropped out and were left with only 38 subjects for their analysis. This is a surprising small group in light of the overall COVID infected population in Thailand and based on a recruitment period from May 2021 to July 2022. They also reported they excluded subjects with depression and anxiety but then report this as an outcome in the recruited cohort. Both the groups original study and this study use healthy individuals as their comparison controls. However relative to their outcomes there is a very large and robust number of studies reporting poor mental health and quality of life outcomes in patients that have been in an ICU and in particular intubated with bacterial and viral non covid infectious pneumonias. The study would had more impact and relevance to at least review briefly and discuss how the results could b compared. Additionally not comparing severity of COVID in an infected cohort but not severe enough to be hospitalized to their cohort raises the important question of whether this less severe cohort could have equivalent outcomes.

The other major concern the reviewer has are the outcomes presented in the manuscript and I will go through each.

It looks like the authors conducted multiple individual T testing instead of a more robust- ANOVA with post hoc analysis? 

Fig 1- after 3 months there was no difference in subjects complaining of one symptom compared to healthy controls. I suspect the SDs were so large to render non significant but these were not shown?

Fig 2- after one month there was no significant difference in either cough or fatigue. In fact I could not find any healthy control expressed data on the fatigue figure?

Fig 3- SF-36 data does show some differences in some of the domains but are these differences actually clinically relevant and how do they compare to a group infected with COVID with less severe complications and compared to a non-COVID ICU cohort.

This is not to diminish the findings and the potential importance of severe COVID and contribution to long term consequences but there are major concerns with the relevance of the reported outcomes 

Author Response

Date 20th June 2023

Editor-in-Chief

Healthcare Journal

Dear Reviewer 2

Subject:  Response to reviewer 2

            On behalf of my co-authors, I am pleased to resubmit this revised Manuscript ID: healthcare-2430660 entitled: “Long-Term Impacts of COVID-19 Pneumonia on Quality of Life” as attached to the Healthcare Journal.

Thank you for your valuable comments.

Comments and Suggestions for Authors

            The authors have attempted to address an important issue related to post covid clinical and mental health consequences (QOL) in their Thai cohort. This appear to be a "follow on" reporting from their original study evaluating very early outcomes. They recruited a very small sample size based on their effects size -power analysis on one month post infection using their pre-existing outcomes based on the EQ -5D-5L survey. They report that 32% of the COVID group dropped out and were left with only 38 subjects for their analysis. This is a surprising small group in light of the overall COVID infected population in Thailand and based on a recruitment period from May 2021 to July 2022. They also reported they excluded subjects with depression and anxiety but then report this as an outcome in the recruited cohort. Both the groups original study and this study use healthy individuals as their comparison controls. However relative to their outcomes there is a very large and robust number of studies reporting poor mental health and quality of life outcomes in patients that have been in an ICU and in particular intubated with bacterial and viral non covid infectious pneumonias. The study would had more impact and relevance to at least review briefly and discuss how the results could b compared. Additionally not comparing severity of COVID in an infected cohort but not severe enough to be hospitalized to their cohort raises the important question of whether this less severe cohort could have equivalent outcomes.

Response: Thank you. We mention them in the limitation of this study. Revised manuscript in the discussion section page no.10, line 300-303, highlighted.

The other major concern the reviewer has are the outcomes presented in the manuscript and I will go through each.

Response: The results section was revised according to your suggestion. Revised manuscript in the results section page no.3-8, highlighted. All figures were also modified per your suggestion. Revised manuscript in the Figure 1-4, page no. 5, 7, and 8. 

It looks like the authors conducted multiple individual T testing instead of a more robust- ANOVA with post hoc analysis?

Response: We re-analyze data according to your suggestion and we added this test used in the statistical analysis part. Revised manuscript in the materials and method section page no.3 line 125-127, highlighted. The p-value from repeated measured ANONA with Bonferroni correction was added in updated table 2 and figure 3. Revised manuscript in the results section page no.6-8.

Fig 1- after 3 months there was no difference in subjects complaining of one symptom compared to healthy controls. I suspect the SDs were so large to render non significant but these were not shown?

Response: The results in Figure 1 were the proportion of subjects had at least one symptom. We added more results in the results section page no.5.

Fig 2- after one month there was no significant difference in either cough or fatigue. In fact I could not find any healthy control expressed data on the fatigue figure?

Response: Because the proportion of subjects with cough and fatigue were zero. We added the label data in the Figure 2. Revised manuscript in the Figure 2, page no.5.

Fig 3- SF-36 data does show some differences in some of the domains but are these differences actually clinically relevant and how do they compare to a group infected with COVID with less severe complications and compared to a non-COVID ICU cohort.

Response: Thank you. These are limitation of our study. We added these limitations in the discussion section. Revised manuscript was highlighted in the discussion section page no. 10, line 300-305.

This is not to diminish the findings and the potential importance of severe COVID and contribution to long term consequences but there are major concerns with the relevance of the reported outcomes.

Response: Thank you. Our aims was to explore the long-term impacts on health-related QoL in post COVID-19 pneumonia patients compared with healthy control subjects with a short interval between visits (months 1, 3, 6, 9, and 12 after hospital discharge).

With regards,

Assoc.Prof. Chalerm Liwsrisakun, MD.

Reviewer 3 Report

Appreciate your efforts in preparing the manuscript.  I have made some comments and suggestions to improve the flow and clarity of the content.  Hope my comments are well received.     

Manuscript ID: healthcare-2430660

TITLE: Long-term impact of COVID-19 Pneumonia on Quality of Life

General Comments: Appreciate the time and the effort preparing the manuscript.  As I reviewed the manuscript, I felt like some sections could benefit clarification and terminology changes.  Hope my comments will be easy to follow.   You also have used “post severe COVID-19 pneumonia” in some sections; please be consistent.  Should this just be post COVID-19 pneumonia?  Some inconsistencies with abbreviations, you should spell them out first and then use the abbreviation throughout the manuscript.  Not sure if you need to say, “Bodily Pain”, just say “pain”; or some sections you have listed “Joint pain”, or added myalgias; make is consistent.

ABSTRACT:

Line #10- Consider revising to: Several studies have reported poor QoL …..Just use “poor” not “poorer”. 

Line#11 – Use (;) between COVID -19 and however.  Or do the following:

Line #11 – Line #12 – This could be a sentence by itself.  However, these studies were limited due to cross-sectional design, longer gap between visits, and lack of controls for comparison.  

Line #13 – Revise to: Therefore, the aim of our prospective study was to assess the impact of COVID-19 pneumonia on QoL, (physical and mental). 

Line #15 – Delete this sentence and start with: We used the 36-item Short form Health Survey (SF-36) and EQ-5D-5L plus Hospital Anxiety and Depression Scale to collect data at months, 1, 3, 6, 9, and 12.

Line #21 – Define EQ-VAS.

Line #22 – Not sure if the word “tended” fits here professionally.  Consider revising to just: improved over one year follow-up period.  Or Improved during the follow-up period.  

INTRODUCTION:

Line #27 – Line #28 – Consider revising for better flow: The COVID-19 infectious respiratory illness caused by SARS-CoV-2 reported since late 2019 lead to a pandemic and has severely impacted human health worldwide.  Not sure if I like my version, but just a thought.

Line # 29 – Line #30 - Delete “there were”, just start the sentence with: Covid-19 infected more than 700 million people and led to over 6-million deaths worldwide till February 2023.  Delete the sentence “This infection is systemic ……not necessary”.

Line #31 – Line #32 – Revise to “Many patients experienced long-term physical and mental health problems called “post COVID-19 condition” or “Long COVID”. 

Line #34 – Line #35 – Line #; Consider revising to: Our previous study reported that nearly one half of patients with post COVID pneumonia still had symptoms including poor QoL and limited exercise capacity compared to the healthy subjects one month after hospital discharge. 

Line #40 – Line # - 45 – Few words can be eliminated: When assessing QoL measured by the SF-36, we discovered that physical, emotional, social, and mental health symptoms were significantly lower in the post COVID-19 group compared to healthy controls. 

Line #45 – Delete “compared and health control groups”. 

Line #48 – Line #53 – Not sure if you need to say p Value < 0.001 ; you just use p < 0.001.   Delete “value”.

Line #56 – Line #58 – Revise to: however, the results from these studies may be limited by the cross-section design, longitudinal and long interval between the visits, and lack of control group. 

MATERIALS AND METHODS:

Line #66 – Delete “also approved”.

Line #68 – Revise to: All subject provided written informed consent prior to enrollment.

Line #69 – Line #Line 79 - We included subjects 18 years of age with the diagnosis of COVID-19 pneumonia with clinical symptoms and evidence of pulmonary infatuation on chest x-ray and a positive RT-PCR between April – May 2021.  We excluded subjects with language barrier (understanding Thai), diagnosis of psychiatric disorders (depression and/or anxiety), COPD, and asthma.  Delete “data collection was conducted in subjects” you have a section on data collection beginning on line #83). 

Line #74 – Line #77- I would suggest moving the location of the study to the top just after your study design. 

Line #86 – Delete the second “history”.

Line #90 – You have already used these abbreviations, not sure why you are spelling them out again.

Line #93 – Change “which is developed” to just “The SF-36 questionnaire developed by”…

Line #95 – Line #96 – Eliminate unnecessary words by just saying “including physical, social mental, emotional, and just general health perception. 

Line #98 – This should say “on a scale of 0 “worst health” to 100 “best health”.

Line 99 – Same here “EQ-5D-5L questionnaire developed by the ….

Line #108 – HADS has already been abbreviated on Line #91, just start the sentence with The HADS questionnaire…..

Line #109 – What do you mean by “during the past week”?  Is this after discharge or just recent episode of depression and/or anxiety?

Line #112 – Consider using “11” indicates ….

Line #116 – Delete “a” just one month after hospital discharge.

Line #117 – Delete “healthy” by now we know control group was the healthy subjects.

Line #120 -Line #124 – The study is complete, so you should past tense.  We calculated the number of participants to be 50 (25 in each group) to be able to reject the null hypothesis….

Line #124 – Change “is” to “was”.

RESULTS:
Line #136 – Revise to: Fifty-six patients from the COVID-19 pneumonia group and 25 patients from the control group participated in the one-month post-hospital discharge follow-up.   

Line #138 - Line #143 – Not sure if “However” is needed.  Just start the sentence with: eighteen post-COVID-19 patients (32.1%) withdrew or dropped out for various reasons (i.e., immigration, loss of contact); therefore, only 38 patients completed the follow-up visits.  Of the 38 remaining patients, 20 were male (52.6%), mean age of 41.1 ± 14.8 years, 22 reported severe pneumonia during admission (……).  

Line #147 – Line #156 – As mentioned above, delete “value” and just report as p=0.016, 0.001, etc.

Line #153 – Revise to: During the study period, 36 patients in the COVID-pneumonia arm and everyone in the control arm received at least one dose of COVID-19 vaccination.   Eight patients in the study arm and five in the control arm had mild COVID-19 infection.

Line #164 – Delete “value”.

Line #166 – The symptoms resolved over time but rebounded in months 9 and 12. 

Line #167 – A minor comment. Change this to a similar trend was observed in patients who only had cough.

Line #177 – Use QoL since it has been abbreviated already.

Line #194 – Change “time Points” to “time frames”.

Line #195 – Revise to “physical and social functioning”….

DISCUSSION:
Line #217 – Line #218 – Delete “s” from impacts;

Line #218 – Not sure if serially can be changed to “successively” or “consecutively” for better wording.

Line #219 – Line #221 - Our study demonstrated that QoL was lower in patients with post COVID-19 pneumonia compared to the control group.  In addition, despite the gradual improvement during the entire follow-up period, it remained lower than the control group.

Line #224 – Change “like the study” to: Similar to the study by Huang et. al. where 68% and 49% ….

Line #225 – Revise to: in our study, cough and fatigue were the most common symptoms presented at a higher rate than in the control group. 

I would suggest some cleaning in this section, you go back and forth talking about getting better to nadir then getting worse.  Just simplify if possible.

Line #228 -Line #229 – Not sure if this sentence is accurate.  I wonder if you are trying to say: the most persistent symptom during later visits was cough.

Line #233 – Change “showed”’ to “reported”.

Line #236 – Delete “the”, just say our study recruited patients….

Line #247 – Change “showed” to “Our study revealed”, also try to clarify “poorer than healthy subject”….

Line # 253 – I think it should be “at 1-month follow-up.

Line #264 -Line #266 – The discrepancy related to the prevalence of depression and anxiety between our study and previous studies may be due to age of the study population.   

Line #275 – Line #278 – Can shorten and simplify to:  The strengths of our study were the prospective design, longitudinal, and the inclusion of control group.   …..

Line #279 – Start with: Our study had several limitations.  First, this was a single center trial.

Line 282 – Delete “were” just say dropped out.

Line #283 – Thirdly, additional confounding factors such as air pollution and the new variant during the 9th month (Jan – Feb) and 12th month visit (April – May 2022) may have influenced our results.

CONCLUSIONS:

Line #289 – Revise to: Among patients with COVID-19 pneumonia, nearly one-third still had symptoms post infection, mainly cough and fatigue. 

Line #292 – Consider replacing “tended” to” expected” to improve over time during 1-year follow-up. 

Line #293 – Line #295 - Consider revising to: Therefore, a multidisciplinary program including physical rehabilitation and psychosocial support is required for managing patients with ongoing complications post COVID-19 infection.  ed on our observation

TABLES:

Table #2 – Consider deleting “in Post COVID-19 Pneumonia” for the table title.  It is in the table and by now the reader knows it is COVID-19 pneumonia and control group. 

FIGURES:
Figure #4 (Line # 212)– Delete “and” between hospital and anxiety at the abbreviation section; it should be Hospital anxiety and depression scale. 

I did see some inconsistencies in abbreviations and few terminologies.  There are few unnecessary wordings and should be eliminated.  

Author Response

Date 20th June 2023

Editor-in-Chief

Healthcare Journal

Dear Reviewer 3

Subject:  Response to reviewer 3

            On behalf of my co-authors, I am pleased to resubmit this revised Manuscript ID: healthcare-2430660 entitled: “Long-Term Impacts of COVID-19 Pneumonia on Quality of Life” as attached to the Healthcare Journal.

Thank you for your valuable comments.

Comments and Suggestions for Authors

            Appreciate your efforts in preparing the manuscript.  I have made some comments and suggestions to improve the flow and clarity of the content.  Hope my comments are well received.    

Manuscript ID: healthcare-2430660

TITLE: Long-term impact of COVID-19 Pneumonia on Quality of Life

General Comments: Appreciate the time and the effort preparing the manuscript.  As I reviewed the manuscript, I felt like some sections could benefit clarification and terminology changes.  Hope my comments will be easy to follow.   You also have used “post severe COVID-19 pneumonia” in some sections; please be consistent.  Should this just be post COVID-19 pneumonia?  Some inconsistencies with abbreviations, you should spell them out first and then use the abbreviation throughout the manuscript.  Not sure if you need to say, “Bodily Pain”, just say “pain”; or some sections you have listed “Joint pain”, or added myalgias; make is consistent.

Response: Thank you. We edited as a whole of manuscript per your suggestion.

ABSTRACT:

Line #10- Consider revising to: Several studies have reported poor QoL …..Just use “poor” not “poorer”.

Response: Thank you. We revised it per your suggestion. Revised manuscript in the abstract section page no.1, line 9, highlighted.

Line#11 – Use (;) between COVID -19 and however.  Or do the following:

Line #11 – Line #12 – This could be a sentence by itself.  However, these studies were limited due to cross-sectional design, longer gap between visits, and lack of controls for comparison. 

Response: Thank you. We separated them for each sentence per your suggestion. Revised manuscript in the abstract section page no.1, line 10-11, highlighted.

Line #13 – Revise to: Therefore, the aim of our prospective study was to assess the impact of COVID-19 pneumonia on QoL, (physical and mental).

Response: Thank you. We revised it per your suggestion. Revised manuscript in the abstract section page no.1, line 11-13, highlighted.

Line #15 – Delete this sentence and start with: We used the 36-item Short form Health Survey (SF-36) and EQ-5D-5L plus Hospital Anxiety and Depression Scale to collect data at months, 1, 3, 6, 9, and 12.

Response: Thank you. We revised it per your suggestion. Revised manuscript in the abstract section page no.1, line 14-16, highlighted.

Line #21 – Define EQ-VAS.

Response: Thank you. We defined EQ-VAS per your suggestion. Revised manuscript in the abstract section page no.1, line 15, highlighted.

Line #22 – Not sure if the word “tended” fits here professionally.  Consider revising to just: improved over one year follow-up period.  Or Improved during the follow-up period. 

Response: Thank you. We revised it per your suggestion. Revised manuscript in the abstract section page no.1, line 21-22, highlighted.

INTRODUCTION:

Line #27 – Line #28 – Consider revising for better flow: The COVID-19 infectious respiratory illness caused by SARS-CoV-2 reported since late 2019 lead to a pandemic and has severely impacted human health worldwide.  Not sure if I like my version, but just a thought.

Response: Thank you. We revised it per your suggestion. Revised manuscript in the introduction section page no.1, line 26-28, highlighted.

Line # 29 – Line #30 - Delete “there were”, just start the sentence with: Covid-19 infected more than 700 million people and led to over 6-million deaths worldwide till February 2023.  Delete the sentence “This infection is systemic ……not necessary”.

Response: Thank you. We revised it per your suggestion. Revised manuscript in the introduction section page no.1, line 28-30, highlighted.

Line #31 – Line #32 – Revise to “Many patients experienced long-term physical and mental health problems called “post COVID-19 condition” or “Long COVID”.

Response: Thank you. We revised it per your suggestion. Revised manuscript in the introduction section page no.1, line 35-36, highlighted.

Line #34 – Line #35 – Line #; Consider revising to: Our previous study reported that nearly one half of patients with post COVID pneumonia still had symptoms including poor QoL and limited exercise capacity compared to the healthy subjects one month after hospital discharge.

Response: Thank you. We revised it per your suggestion. Revised manuscript in the introduction section page no.1, line 36-39, highlighted.

Line #40 – Line # - 45 – Few words can be eliminated: When assessing QoL measured by the SF-36, we discovered that physical, emotional, social, and mental health symptoms were significantly lower in the post COVID-19 group compared to healthy controls.

Response: Thank you. We revised it per your suggestion. Revised manuscript in the introduction section page no.1, line 42-45, highlighted.

Line #45 – Delete “compared and health control groups”.

Response: Thank you. We deleted it per your suggestion. Revised manuscript in the introduction section page no.1 , line 44-45, highlighted.

Line #48 – Line #53 – Not sure if you need to say p Value < 0.001 ; you just use p < 0.001.   Delete “value”.

Response: Thank you. But, because of the comments from another reviewer to remove numerical and statistical data from this section, we have to delete all p Value in the introduction section. However, according to MDPI format, we confirm to use p Value in other parts of this manuscript.

Line #56 – Line #58 – Revise to: however, the results from these studies may be limited by the cross-section design, longitudinal and long interval between the visits, and lack of control group.

Response: Thank you. We revised it per your suggestion. Revised manuscript in the introduction section page no.2, line 53-55, highlighted.

MATERIALS AND METHODS:

Line #66 – Delete “also approved”.

Response: Thank you. We deleted it per your suggestion. Revised manuscript in the materials and methods section page no.2, line 66, highlighted.

Line #68 – Revise to: All subject provided written informed consent prior to enrollment.

Response: Thank you. We revised it per your suggestion. Revised manuscript in the materials and methods section page no.2 , line 67-68, highlighted.

Line #69 – Line #Line 79 - We included subjects ≥ 18 years of age with the diagnosis of COVID-19 pneumonia with clinical symptoms and evidence of pulmonary infatuation on chest x-ray and a positive RT-PCR between April – May 2021.  We excluded subjects with language barrier (understanding Thai), diagnosis of psychiatric disorders (depression and/or anxiety), COPD, and asthma.  Delete “data collection was conducted in subjects” you have a section on data collection beginning on line #83).

Response: Thank you. We revised it per your suggestion. Revised manuscript in the materials and methods section page no.2, line 68-75, highlighted.

Line #74 – Line #77- I would suggest moving the location of the study to the top just after your study design.

Response: Thank you. We revised it per your suggestion. Revised manuscript in the materials and methods section page no.2, line 61-63, highlighted.

Line #86 – Delete the second “history”.

Response: Thank you. We deleted it per your suggestion. Revised manuscript in the materials and methods section page no.2, line 81, highlighted.

Line #90 – You have already used these abbreviations, not sure why you are spelling them out again.

Response: Thank you. We revised it per your suggestion. Revised manuscript in the materials and methods section page no.2, line 84-85, highlighted.

Line #93 – Change “which is developed” to just “The SF-36 questionnaire developed by”…

Response: Thank you. We revised it per your suggestion. Revised manuscript in the materials and methods section page no.2 line 87-88, highlighted.

Line #95 – Line #96 – Eliminate unnecessary words by just saying “including physical, social mental, emotional, and just general health perception.

Response: Thank you. We revised it per your suggestion. Revised manuscript in the materials and methods section page no.2, line 87-91, highlighted.

Line #98 – This should say “on a scale of 0 “worst health” to 100 “best health”.

Response: Thank you. We revised it per your suggestion. Revised manuscript in the materials and methods section page no.2, line 90-91, highlighted.

Line 99 – Same here “EQ-5D-5L questionnaire developed by the ….

Response: Thank you. We revised it per your suggestion. Revised manuscript in the materials and methods section page no.2, line 92-93, highlighted.

Line #108 – HADS has already been abbreviated on Line #91, just start the sentence with The HADS questionnaire…..

Response: Thank you. We revised it per your suggestion. Revised manuscript in the materials and methods section page no.3, line 101-102, highlighted.

Line #109 – What do you mean by “during the past week”?  Is this after discharge or just recent episode of depression and/or anxiety?

Response: Thank you. It is one week prior to questionnaire response. We used “one week prior to questionnaire response” instead of “during the past week” for clarity. Revised manuscript in the materials and methods section page no.3, line 102, highlighted.

Line #112 – Consider using “≥11” indicates ….

Response: Thank you. We revised it per your suggestion. Revised manuscript in the materials and methods section page no.3, line 105, highlighted.

Line #116 – Delete “a” just one month after hospital discharge.

Response: Thank you. We revised it per your suggestion. Revised manuscript in the materials and methods section page no.3, line 111, highlighted.

Line #117 – Delete “healthy” by now we know control group was the healthy subjects.

Response: Thank you. We revised it per your suggestion. Revised manuscript in the materials and methods section page no.3, line 112, highlighted.

Line #120 -Line #124 – The study is complete, so you should past tense.  We calculated the number of participants to be 50 (25 in each group) to be able to reject the null hypothesis….

Response: Thank you. We revised it per your suggestion. Revised manuscript in the materials and methods section page no.3, line 115-118, highlighted.

Line #124 – Change “is” to “was”.

Response: Thank you. We changed it per your suggestion. Revised manuscript in the materials and methods section page no.3, line 118, highlighted.

RESULTS:

Line #136 – Revise to: Fifty-six patients from the COVID-19 pneumonia group and 25 patients from the control group participated in the one-month post-hospital discharge follow-up.  

Response: Thank you. We revised it per your suggestion. Revised manuscript in the results section page no.3, line 131-132, highlighted.

Line #138 - Line #143 – Not sure if “However” is needed.  Just start the sentence with: eighteen post-COVID-19 patients (32.1%) withdrew or dropped out for various reasons (i.e., immigration, loss of contact); therefore, only 38 patients completed the follow-up visits.  Of the 38 remaining patients, 20 were male (52.6%), mean age of 41.1 ± 14.8 years, 22 reported severe pneumonia during admission (……). 

Response: Thank you. We revised it per your suggestion. Revised manuscript in the results section page no.3, line133-138, highlighted.

Line #147 – Line #156 – As mentioned above, delete “value” and just report as p=0.016, 0.001, etc.

Response: Thank you. According to MDPI format. We confirm to use p Value.

Line #153 – Revise to: During the study period, 36 patients in the COVID-pneumonia arm and everyone in the control arm received at least one dose of COVID-19 vaccination.   Eight patients in the study arm and five in the control arm had mild COVID-19 infection.

Response: Thank you. We revised it per your suggestion. Revised manuscript in the results section page no.4, line 149-152, highlighted.

Line #164 – Delete “value”.

Reply: Thank you. According to MDPI format. We confirm to use p Value.

Line #166 – The symptoms resolved over time but rebounded in months 9 and 12.

Response: Thank you. We revised it per your suggestion. Revised manuscript in the results section page no.4, line 164-165, highlighted.

Line #167 – A minor comment. Change this to a similar trend was observed in patients who only had cough.

Response: Thank you. We revised it per your suggestion. Revised manuscript in the results section page no.4, line 165-166, highlighted.

Line #177 – Use QoL since it has been abbreviated already.

Response: Thank you. We revised it per your suggestion. Revised manuscript in the results section page no.5, line 174, highlighted.

Line #194 – Change “time Points” to “time frames”.

Response: Thank you. However, we already revised this sentence. Revised manuscript in the results section page no.6, line 193-195, highlighted.

Line #195 – Revise to “physical and social functioning”….

Response: Thank you. We revised it per your suggestion. Revised manuscript in the results section page no.6, line 193-194, highlighted.

DISCUSSION:

Line #217 – Line #218 – Delete “s” from impacts;

Response: Thank you. We deleted it per your suggestion. Revised manuscript in the discussion section page no.8, line 219, highlighted.

Line #218 – Not sure if serially can be changed to “successively” or “consecutively” for better wording.

Response: Thank you. We changed it per your suggestion. Revised manuscript in the discussion section page no.8, line 220, highlighted.

Line #219 – Line #221 - Our study demonstrated that QoL was lower in patients with post COVID-19 pneumonia compared to the control group.  In addition, despite the gradual improvement during the entire follow-up period, it remained lower than the control group.

Response: Thank you. We revised it per your suggestion. Revised manuscript in the discussion section page no.8-9, line 221-224, highlighted.

Line #224 – Change “like the study” to: Similar to the study by Huang et. al. where 68% and 49% ….

Response: Thank you. We revised it per your suggestion. Revised manuscript in the discussion section page no.9, line 226-227, highlighted.

Line #225 – Revise to: in our study, cough and fatigue were the most common symptoms presented at a higher rate than in the control group.

Response: Thank you for your suggestion. However, we have to revise this paragraph of manuscript per comments by other reviewer in the discussion section page no.9, line 226-243, highlighted.

I would suggest some cleaning in this section, you go back and forth talking about getting better to nadir then getting worse.  Just simplify if possible.

Response: Thank you for your suggestion. However, we have to revise this paragraph of manuscript per comments by other reviewer in the discussion section page no.9, line 226-243, highlighted.

Line #228 -Line #229 – Not sure if this sentence is accurate.  I wonder if you are trying to say: the most persistent symptom during later visits was cough.

Response: Thank you for your suggestion. However, we have to revise this paragraph of manuscript per comments by other reviewer in the discussion section page no.9, line 226-243, highlighted.

Line #233 – Change “showed”’ to “reported”.

Response: Thank you. We changed it per your suggestion. Revised manuscript in the discussion section page no.9, line 245, highlighted.

Line #236 – Delete “the”, just say our study recruited patients….

Response: Thank you. We deleted it per your suggestion. Revised manuscript in the discussion section page no. 9, line 248-249, highlighted.

Line #247 – Change “showed” to “Our study revealed”, also try to clarify “poorer than healthy subject”….

Response: Thank you. We changed it per your suggestion. Revised manuscript in the discussion section page no.9, line 260-263, highlighted.

Line # 253 – I think it should be “at 1-month follow-up.

Response: Thank you. We revised it per your suggestion. Revised manuscript in the discussion section page no.9, line 271, highlighted.

Line #264 -Line #266 – The discrepancy related to the prevalence of depression and anxiety between our study and previous studies may be due to age of the study population.  

Response: Thank you. We revised it per your suggestion. Revised manuscript in the discussion section page no.9, line 282-284, highlighted.

Line #275 – Line #278 – Can shorten and simplify to:  The strengths of our study were the prospective design, longitudinal, and the inclusion of control group.   …..

Response: Thank you. We revised it per your suggestion. Revised manuscript in the discussion section page no.10, line 293-294, highlighted.

Line #279 – Start with: Our study had several limitations.  First, this was a single center trial.

Response: Thank you. We revised it per your suggestion. Revised manuscript in the discussion section page no.10, line 294-295, highlighted.

Line 282 – Delete “were” just say dropped out.

Response: Thank you. We deleted “were” per your suggestion. Revised manuscript in the discussion section page no.10, line 296, highlighted.

Line #283 – Thirdly, additional confounding factors such as air pollution and the new variant during the 9th month (Jan – Feb) and 12th month visit (April – May 2022) may have influenced our results.

Response: Thank you. We revised it per your suggestion. Revised manuscript in the discussion section page no.10, line 297-300, highlighted.

CONCLUSIONS:

Line #289 – Revise to: Among patients with COVID-19 pneumonia, nearly one-third still had symptoms post infection, mainly cough and fatigue.

Response: Thank you. We revised it per your suggestion. Revised manuscript in the conclusions section page no.10, line 308-309, highlighted.

Line #292 – Consider replacing “tended” to” expected” to improve over time during 1-year follow-up.

Response: Thank you. We changed it per your suggestion. Revised manuscript in the conclusion section page no.10, line 309-310, highlighted.

Line #293 – Line #295 - Consider revising to: Therefore, a multidisciplinary program including physical rehabilitation and psychosocial support is required for managing patients with ongoing complications post COVID-19 infection.  ed on our observation

Response: Thank you. We revised it per your suggestion. Revised manuscript in the conclusions section page no.10, line 311-313, highlighted.

TABLES:

Table #2 – Consider deleting “in Post COVID-19 Pneumonia” for the table title.  It is in the table and by now the reader knows it is COVID-19 pneumonia and control group.

Response: Thank you. We deleted it per your suggestion. Revised manuscript in the Table 2 page no.6, line 186, highlighted.

FIGURES:

Figure #4 (Line # 212)– Delete “and” between hospital and anxiety at the abbreviation section; it should be Hospital anxiety and depression scale.

Response: Thank you. We revised it per your suggestion. Revised manuscript in the Figure 4 page no.8, line 214-215, highlighted.

Comments on the Quality of English Language

            I did see some inconsistencies in abbreviations and few terminologies.  There are few unnecessary wordings and should be eliminated. 

Response: Thank you for your suggestion. We revised as a whole of manuscript per your suggestion.

With regards,

Assoc.Prof. Chalerm Liwsrisakun, MD.

Round 2

Reviewer 2 Report

Thank you for addressing the limitations of your paper in an important area. I have reviewed your revision and find them very acceptable  

Author Response

Thank you for your kindly response.